# A Systematic Review of Sleep in Patients with Disorders of Consciousness: From Diagnosis to Prognosis

**DOI:** 10.3390/brainsci11081072

**Published:** 2021-08-16

**Authors:** Jiahui Pan, Jianhui Wu, Jie Liu, Jiawu Wu, Fei Wang

**Affiliations:** 1School of Software, South China Normal University, Foshan 528225, China; 2020023817@m.scnu.edu.cn (J.W.); 20172005084@m.scnu.edu.cn (J.L.); 20182005240@m.scnu.edu.cn (J.W.); fwang@scnu.edu.cn (F.W.); 2Pazhou Lab, Guangzhou 510330, China

**Keywords:** sleep, electroencephalography (EEG), disorder of consciousness (DOC), minimally conscious state (MCS), unresponsive wakefulness syndrome (UWS)

## Abstract

With the development of intensive care technology, the number of patients who survive acute severe brain injury has increased significantly. At present, it is difficult to diagnose the patients with disorders of consciousness (DOCs) because motor responses in these patients may be very limited and inconsistent. Electrophysiological criteria, such as event-related potentials or motor imagery, have also been studied to establish a diagnosis and prognosis based on command-following or active paradigms. However, the use of such task-based techniques in DOC patients is methodologically complex and requires careful analysis and interpretation. The present paper focuses on the analysis of sleep patterns for the evaluation of DOC and its relationships with diagnosis and prognosis outcomes. We discuss the concepts of sleep patterns in patients suffering from DOC, identification of this challenging population, and the prognostic value of sleep. The available literature on individuals in an unresponsive wakefulness syndrome (UWS) or minimally conscious state (MCS) following traumatic or nontraumatic severe brain injury is reviewed. We can distinguish patients with different levels of consciousness by studying sleep patients with DOC. Most MCS patients have sleep and wake alternations, sleep spindles and rapid eye movement (REM) sleep, while UWS patients have few EEG changes. A large number of sleep spindles and organized sleep–wake patterns predict better clinical outcomes. It is expected that this review will promote our understanding of sleep EEG in DOC.

## 1. Introduction

Due to advances in critical care, an increasing number of patients survive acute brain injury, causing an increased incidence and prevalence of patients with disorders of consciousness (DOC). DOC encompasses coma, vegetative state (VS)/unresponsive wakefulness syndrome (UWS), and minimally conscious state (MCS). Patients with UWS have a sleep–wake cycle, but they completely lose their awareness of themselves and their surroundings [1], while patients with MCS have awareness and show purposeful behaviors but are unable to communicate effectively [2]. Recently, MCS has been further divided into two substates, MCS+ (high-level behavioral responses, such as command following) and MCS− (low-level behavioral responses, such as visual pursuit and pain localization) [3]. The gold standard Coma Recovery Scale-Revised (CRS-R) is the best behavioral assessment criterion for the diagnosis of UWS or MCS, but patients who are unable to follow the commands due to motor impairments may receive an incorrect diagnosis of UWS. Therefore, the misdiagnosis rate has been reported to be as high as 43% [4].

Electroencephalography (EEG) is a noninvasive, safe, and relatively convenient technique to record brain activities, which allows quantitative methods to detect changes and patterns of EEG signals related to DOC [5,6]. The best feature of EEG data is neural oscillations [7]. From the perspective of biophysics, EEGs are extracellular currents that reflect the total dendritic postsynaptic potentials in millions of parallel pyramidal cells [8]. Certain characteristics of the EEG are signs of corticothalamic integrity, which is considered to be the main basis of wakeful consciousness [9]. Several studies have reported the informative value of evoked potentials in task-based or active paradigms during the awake state [10,11,12,13]. However, patients with DOC are easily fatigued and have a considerably limited attention span, which results in false-negative findings. Compared with these techniques, sleep assessment could provide an alternative way to assess residual brain function to refine diagnosis and prognosis in DOC [14]. First, many cognitive functions (e.g., language understanding, stimuli selection, and sustained attention) are not required for EEG recording during the sleep state. Second, close relationships between the quality of neurophysiological sleep patterns and clinical symptoms have been demonstrated in a number of neurological diseases. Many highly reliable electrophysiological features can be observed during sleep, such as spindles, K-complex, slow waves, and rapid eye movements. These features could be observed and accounted for in long-term monitoring using EEG or polysomnography (PSG). Third, sleep EEG can minimize subjectivity and human errors in both diagnosis and prognosis.

From a neurobiological viewpoint, consciousness and sleep are intimately linked [14]. Regular sleep patterns could reflect the preservation of brain functions [15] and play a key role in memory consolidation [16], hormonal regulation [17], and immune functions [18]. A better understanding of sleep-in patients with DOC could be of great help in distinguishing patients with different levels of consciousness. Subsequent findings further demonstrated that the integrity of identified sleep patterns carries important prognostic information for outcomes of consciousness recovery [19]. Consequently, the diagnostic and prognostic value of sleep in DOC has received increasing attention.

Neurophysiological changes in sleep have been well studied in healthy humans [20]. However, more detailed sleep assessment for DOC patients is still a controversial issue. Some research groups believe that manual sleep staging is feasible [21,22], while others [23] believe that it is impossible to perform sleep staging according to the established criteria of the American Academy of Sleep Medicine (AASM) or Rechtschaffen and Kales (R&K). Furthermore, the frequency, topography, power, and shape of EEG or PSG signals are changed in patients with severe brain injury. It is difficult to find polysomnographic patterns, such as sleep spindles, K-complex, and rapid eye movement.

Considering the above factors, we present a review of studies on sleep EEG for patients with DOC in this paper. First, the progress of sleep stage classification in patients with DOC is introduced. Second, the diagnostic methods of patients with DOC using sleep EEG are mainly described. Third, the prognostic method and value of sleep in DOC patients were analyzed. Furthermore, the current challenges and future prospects of sleep EEG in DOC are summarized and discussed. It is hoped that these resources can improve our knowledge of sleep EEG patterns in the evaluation, diagnosis, and prognosis in cases of DOC, provide some ideas and reduce obstacles for clinical rehabilitation.

## 2. An Overview of Sleep EEG in Patients with DOC

This systematic review was conducted according to PRISMA guidelines [24]. As shown in Figure 1, we searched the PubMed database using the concepts of sleep and DOC. Search words included ((disorders of consciousness) OR (DOC)) AND (sleep)), and the field search was (title/abstract). The number of journal papers found from 2002–2021 was 716. We emphasized the articles published in the last 20 years regarding the diagnosis and classification of DOC without language restrictions. Exclusion (*n* = 649) were records not closely related to the classification, diagnosis, and prognosis of DOC, which mainly included the studies of the state of consciousness under sleep or anesthesia, the studies of sleep under different lifestyles, the studies of the clinical manifestations of patients with schizophrenia, Alzheimer’s disease, or other mental disorders using different drugs, and the studies involving other types of measurements, such as functional magnetic resonance imaging (fMRI), positron emission tomography (PET) or transcranial magnetic stimulation (TMS), or duplicates. The final result was that 36 articles were included. We focused on articles using sleep EEG or PSG methods, as well as studies on patients diagnosed with coma, VS/UWS, or MCS. We believe that it is a good time to summarize new technologies and address the gap between theory and application in this field.

Studies related to the topic of sleep EEG in patients with DOC can be devised into three main classes. First, each sleep state is characterized by a different type of EEG activity, and thus EEG analysis has been used for sleep stage classification. Second, using typical sleep EEG waveforms for detection and diagnosis in patients with DOC has been reported by some researchers [21,24,25]. Third, sleep EEG changes have a predictive value in patients with DOC. The state-of-the-art of the above three classes of sleep EEG are reviewed in the following sections.

## 3. Sleep Stage Classification in Patients with DOC

PSG is the main tool to evaluate sleep in the laboratory and can be used for clinical and research purposes. PSG is used to collect EEG, EOG, EMG, electrocardiogram, pulse oximetry, airflow, and respiratory effort during sleep and utilize these to evaluate the underlying causes of sleep disturbances [26]. PSG can provide much information about the integrity of the global brain network; thus, sleep assessment can contribute to the diagnosis of DOC patients. During PSG monitoring, EEG and other sensors are used to divide sleep into several distinct stages. Sleep can be roughly divided into NREM sleep and REM sleep. The sleep stages cycles from NREM sleep stage 1 (N1) to REM sleep and then starts again from stage N1. A complete sleep cycle takes approximately 90 to 110 min, and each stage lasts 5 to 15 min. Figure 2 describes the distinct sleep stages of healthy subjects for more than 8 h. The sleep staging process can be very complicated. Many parameters of sleep staging need to be taken into consideration at the same time, and the contextual epoch scores also need to be considered. Compared with records from healthy subjects, scoring records from subjects with specific sleep disorders can be more challenging.

### 3.1. Sleep Stage

In 2007, the new AASM Manual for the Scoring of Sleep and Associated Events was published, which provides a comprehensive reference for the assessment of PSG data. According to the new standard, the following staging system was proposed. AASM sleep staging is divided into five stages, including stage W (wakefulness), stage R (REM sleep), N1 stage (non-REM sleep stage 1), N2 stage (non-REM sleep stage 2), and N3 (non-REM sleep stage 3). Unlike the sleep staging of the R&K standard published in 1968, the new standard has incorporated stage 3 sleep (S3) and stage 4 sleep (S4) into N3. The characteristics of the sleep stages are summarized as follows.

The characteristic of stage W is the appearance of an alpha rhythm in the EEG signal. Theta waves can be observed in stage N1. Sleep spindles and K-complexes may be detected in stage N2. Stage N3 is the deep sleep stage of sleep, in which slow waves and delta waves are dominant in the EEG signal. In addition, spindles may occur at this stage. In stage REM, rapid eye movements occur, and there is no obvious characteristic in the EEG signal. Theta waves and possible sawtooth waves in the EEG signal can be observed [27].

### 3.2. Methods in Diagnosis of DOC

Sleep parameters such as sleep spindle, slow wave sleep, and rapid eye movement sleep can be used as independent markers of the severity of consciousness impairment. The purpose of sleep staging is to identify the stages of sleep that are crucial in the diagnosis and treatment of sleep disorders. The principle of sleep staging is to divide the night into continuous periods of 30 s, called epochs. Traditionally, doctors use these epochs to assess and analyze sleep patterns. However, the traditional manual sleep staging method is time-consuming and relies on the experience of doctors. Therefore, automatic sleep staging methods have become very important in recent years.

There are many studies in automatic sleep stage classification methods based on multiple signals such as EEG, EOG, and EMG [28,29,30], or single-channel EEG [31,32,33]. The classification goal is often accomplished by statistical rules and deep learning technology. The former focuses on the selection of features and classifiers, which does not require much training data, while the latter focuses on neural network input and structure, which has high requirements for the quality and quantity of training data. Conventional machine learning methods for automatic sleep stage classification include naive Bayes [34], support vector machines [35], and random forests [36]. Recently, a large number of deep learning methods have been employed in automatic sleep staging. At present, there are several depth network structures for sleep stage scoring, such as convolutional neural networks (CNNs) [37] and recurrent neural networks (RNNs) [29]. In our previous studies [38,39], an automatic sleep staging method based on ICA-ReliefF was proposed on the Sleep-EDF database. The overall accuracy of the Sleep-EDF database reached 90.10 ± 2.68%, and the kappa coefficient was 0.87 ± 0.04.

## 4. Diagnosis of Patients with DOC Using Sleep EEG

The ability to distinguish an MCS from a UWS offers crucial value for family counseling, treatment decision-making and rehabilitation plan design. Currently, behavior-based CRS-R assessment is predominantly used in the diagnosis of patients with DOC. Some reliable characterization of sleep may help us understand the DOC patient’s pathological conditions and improve the diagnosis and prognosis, but a standardized sleep assessment procedure has not yet been established. Some attempts [21,24,25] automatically assess sleep architecture in patients with DOC, while several studies [1,2,40] have shown the potential of sleep-like activity on EEG in detecting residual cognition functions in patients with DOC. Wielek et al. [21] recently proposed a novel data-driven method and used machine learning techniques to analyze quantitative EEG signals. The long-term PSG recordings in day and night periods for 23 DOC patients were classified into one of the five sleep stages (i.e., wake, N1, N2, N3, or R), providing new insights into sleep patterns and brain functions of DOC patients. Therefore, one of the primary applications of sleep EEG studies in patients with DOC is auxiliary diagnosis. The relation between sleep pattern and cognitive function is summarized in Figure 3.

As shown in Figure 3, subjects in MCS and Normal showed all stages of sleep in contrast to UWS patients. This indicates that the brain function of these patients has been fully protected [41]. MCS patients exhibit relatively preserved thalamocortical connectivity compared to UWS patients. Compared to MCS patients, the UWS patients did not show homoeostatic regulation, a detectable sleep cycle or slow wave activity. Since slow wave activity is considered to be related to plasticity [42], the better prognosis observed in the studies comparing patients with MCS to those with UWS can also be partially explained in their findings [43]. S. De Salvo et al. [44] proposed a system named Neurowave to monitor event-related potentials (ERPs) evoked by neurosensory stimulation in 11 VS and 5 MCS patients. The absence of an ERP component could be a distinctive marker between VS and MCS patients. The other differences in sleep elements between UWS and MCS patients are summarized in Table 1.

### 4.1. Sleep–Wake Cycle

The human circadian rhythm is controlled by the suprachiasmatic nucleus in the brain. At the behavioral level, the most famous cycle of circadian rhythm (i.e., sleep–wake rhythm) can be observed at the level of arousal with changes in heart rate, blood pressure, hormone secretion, or body temperature.

The emergence of the eye-opening periods, the reappearance of the circadian rhythm and the behavioral sleep–wake cycles prove that DOC patients come from a coma to MCS [48]. Recently, Blume et al. [49] reported that the integrity of patients’ circadian rhythms was associated with arousal levels, possibly due to better depiction of periods of sleep and wakefulness. By observing the behavior of patients with DOC for prolonged eye-opening or eye-closing periods, it can be inferred whether there is a circadian rhythm sleep–wake cycle in DOC patients. However, although the existence of a sleep–wake cycle is important for differential diagnosis, there is little evidence that patients with DOC exhibit circadian rhythms or sleep–wake cycles similar to those of healthy people. Past studies [50,51] have reported differences in the circadian activity of most patients with UWS and MCS, and the signs of circadian rhythm in MCS patients are more pronounced.

Studies [52] have shown that UWS patients exhibit sleep–wake cycles. The sleep-related behavioral characteristics of UWS patients are similar to those of normal individuals. During recovery from UWS, these findings may have an impact on the restructuring assessment of rapid eye movement sleep [52]. Landsness et al. [2] observed the behavioral sleep pattern of five UWS patients, but no electrophysiological sleep–wake pattern, while the sleep pattern of six MCS patients was close to normal. They believe that the nocturnal electrophysiological sleep characteristics observed from these results may be a reliable indicator of the patient’s consciousness level, distinguishing UWS from MCS. The presence of a normal sleep pattern is related to the level of consciousness related to clinical and neuroscience. When evaluating brain function in patients with noncommunication brain injury, such studies may be a useful supplement to bedside behavioral assessment [53].

### 4.2. Rapid Eye Movement Sleep and Slow-Wave Sleep

The presence of rapid eye movement sleep and slow oscillations may indicate that the function of the pontine tegmentum in the brainstem is preserved and that the function of certain thalamic cortical rings and brainstem nuclei is intact [54,55]. Others have suggested that the number of sleep spindles or the shape of slow oscillations in DOC patients may reflect the preservation of the thalamocortical system and even the state of consciousness [1].

Oksenberg et al. [56] found that the phasic activities of rapid eye movement sleep in 11 UWS patients were significantly reduced, but the number of these activities had nothing to do with the recovery of the clinical condition because experiments indicated that there was no obvious difference in rapid eye movement sleep phasic activities between the patients who recovered consciousness and those who did not. UWS is caused by overwhelming damage to the cerebral hemisphere, resulting in a large loss of cortical activity but retaining the functional brain stem, allowing continuous regulation of primitive reflexes and vegetative functions. Although UWS retains the brain stem mechanism responsible for the sleep wake cycle and the emergence of rapid eye movement sleep, the significant decrease in rapid eye movement sleep phase activity indicates that other brain stem mechanisms are impaired [56].

### 4.3. Sleep Spindles

Sleep spindles are relatively lacking in patients with DOC, and the abnormalities of sleep spindles in patients with UWS are greater than those in MCS patients. A few UWS patients and most MCS patients had preserved spindles, SWS, and rapid eye movement sleep [1]. In one study [40], Forgacs et al. described the characteristics of 44 DOC patients with conventional EEG. In approximately one-third of UWS patients and in more than half of MCS patients, preserved sleep spindles, rapid eye movement, and slow-wave sleep can be seen. Moreover, the presence, quality and quantity of sleep spindles in both patients with UWS and MCS were associated with more favorable outcomes. In addition, based on their research findings, UWS or MCS patients who have severely abnormal EEG background activity will be less likely to have a high level of cognitive functions demonstrated by functional neuroimaging.

## 5. Prognostic Value of Sleep

Under the current clinical conditions, the prognosis for survival and recovery of DOC is still difficult. First, sleep characteristics, such as spindles, SWS, and rapid eye movement, in patients with DOC are different from those in healthy individuals in many respects. Second, EEG or PSG recording in the clinic is difficult because the clinical instability and nursing activities of patients with DOC often result in artifacts.

Currently, the factors of sleep that can influence or assist in prognosis can be roughly summarized into sleep–wake patterns, sleep spindles, and environmental factors, as shown in Figure 4. The existence of sleep cycles, organized sleep patterns and the homoeostatic regulation of slow waves are common features in healthy individuals. The integrity of the global brain could also be reflected by the presence of standard sleep elements, as these elements have been confirmed to be altered in several pathological states, such as Alzheimer’s disease [57] and stroke [58].

The presence of EEG patterns similar to sleep may be a predictor of a favorable outcome. According to reports, patients’ sleep patterns will become more complex as cognition gradually recovers during rehabilitation therapy. The possible relationship of activity to patient outcome is summarized in Table 2. Pavlov and his colleague have shown that these sleep parameters can predict whether DOC patients can regain consciousness [59]. Scarpino et al. [60] found that the use of EEG in patients with severe DOC after acquired brain injuries could improve the neurological prognosis when discharged from the intensive rehabilitation unit. Multivariable analysis showed that specific EEG patterns were independent predictors of improvement in consciousness when UWS patients were discharged from the hospital. They also proposed an EEG score based on the terms of the American Clinical Neurophysiology Society (ACNS), which showed the highest accuracy of good neurological prognosis in patients with severe DOC after the acute phase [61].

Prognosis in patients with DOC depends primarily on etiology, age and the time interval after brain injury, as well as the Glasgow Coma Scale (GCS) subscores, EEG signals and some sensory evoked potentials of the patient. Studies on sleep in subjects with DOC suggested that the retained functional integrity of the thalamus may be reflected by spindle waves, the residual functioning of brainstem nuclei may be reflected by SWS and rapid eye movement sleep, and the circadian organization of sleep patterns can provide information about residual hypothalamic functioning. All these issues are discussed in the following.

### 5.1. Standard Spindles

The amount and characteristics of sleep spindle waves may be helpful in distinguishing patients with MCS and UWS and their early prognosis [47,48]. In a recent systematic review of brain measurements in DOC patients [58], the authors reported that the stimulation-induced vibration EEG responses could be used as a predictor of outcomes. They further emphasized that their prognostic value of sleep spindles is promising.

Cologan et al. [1] found that the existence of standard spindles often predicted better clinical outcomes. In their study, 20 patients who were in a UWS (*n* = 10) or in an MCS (*n* = 10) with brain damage underwent 24-h polysomnography. Standard spindles were visible in 4/10 UWS and 7/10 MCS patients. Six of the seven patients showed standard spindles in their recording with a favorable outcome. Eight of the 13 patients had no spindles, and the other five patients showed a small number of standard spindles (*n* < 10) with an unfavorable outcome.

### 5.2. Organized Sleep–Wake Patterns

Valente et al. [62] found that 86% of the 24 patients with well-structured NREM and/or REM sleep elements had no sequelae or that only a small amount of neurological deficits showed good recovery through experiments. In a subsequent study, Arnaldi et al. [19] compared the potential prognostic value of sleep/wake patterns in subacute DOCs using 24-h PSG. They suggested that persistent and more organized sleep patterns might be reliable predictors of positive outcomes in subacute DOC patients. These characteristics may be more convincing than the existing prognostic factors, such as patient age and clinical status. In the field of neurorehabilitation, obtaining reliable prognostic data could help optimize the treatment of patients with severe brain injury and assess the prognosis of patients. Compared with other classic parameters (e.g., GCS or neuroimaging), the organization of sleep patterns is a better predictor of the prognosis for survival and functional recovery. The preservation of the cycle of NREM and REM sleep means a better functional integrity of the CNS. The data in [62] showed that the sleep pattern detected by 24-h PSG monitoring after the end of the acute phase might be a very reliable prognostic indicator in head injury coma. In fact, a more complete sleep structure may mean a higher level of consciousness. Therefore, a poor sleep structure often represents a bad outcome [2,54].

### 5.3. Factors of Sleep Abnormalities

Sleep is closely related to factors—such as social pressure, age, physical illness, and mental state—while DOC patients are drastically different from healthy individuals in these factors, which makes it intractable to interpret sleep abnormalities in DOC. Even at night, DOC patients are regularly disturbed by light, sounds and detection equipment in the hospital setting. They will also be periodically awakened and moved by personnel to avoid decubitus ulcers, which may affect sleep. Moreover, patients with DOC do not encounter the social pressure that healthy individuals usually have, and they can sleep almost as long as they want, so there is less need for night sleep. These are just a few external factors that interfere with sleep in addition to the internal factors associated with brain injury. Future 24-h monitoring research should focus more specifically on the distribution of different stages and phases of the daily cycle so that we can control the nighttime sleep abnormalities caused by daytime naps [22].

## 6. Future Challenges and Directions

This paper focuses on sleep stage classification, identification and prediction of sleep-in patients with DOC. First, we described the progress of the sleep staging method in healthy subjects and DOC patients. The classification of sleep/wake periods by the data-driven approach overcomes the existing strong subjective and rough evaluation of DOC patients’ shortcomings. Next, we mainly analyzed studies about the identification of patients with DOC using sleep EEG (in Table 1). Circadian rhythms, sleep–wake cycle, spindles, SWS, and rapid eye movement sleep have been used to differentiate patients with MCS from those with UWS. Furthermore, the possible relationship of sleep activity and patient outcome that are frequently used in DOC studies is reviewed and summarized (in Table 2). The retained functional integrity of the thalamus may be reflected by spindle waves, the residual functioning of brainstem nuclei may be reflected by SWS and rapid eye movement sleep, and the circadian organization of sleep patterns can provide information about residual hypothalamic functioning. In the following, we consider some challenges and directions of sleep EEG in DOC patients in future studies.

### 6.1. PSG Recordings in DOC

In addition, it is challenging to record high-quality PSG signals in DOC patients as a result of electrical artifacts caused by thermal dysregulation, strong sweating, skin, and skull lesions or medical equipment. To obtain clean EEG data for in-depth analysis and scientific data explanation, sophisticated correction methods such as all kinds of independent component analysis algorithms may be necessary. Another issue is that 24 h is the minimal but not the best recording time. For some patients, the observation they receive may eventually be an atypical day. However, it would bring a considerable economic and technical challenge while adopting 48- or 72-h PSG recordings.

### 6.2. Sleep Scoring Rules in Patients with DOC

From the perspective of electrophysiology, we learn little about UWS or MCS sleep. As conventional sleep scoring rules are hardly making available for patients with DOC, the presence of sleep-specific PSG graphoelements and the classification of sleep is still a matter of scientific controversy. The presence and characteristics of sleep in DOC patients seem to be the most challenging problem as they do not show normal physiological, behavioral, and regulatory signs of sleep. Conventional sleep criteria (e.g., R&K) have been applied to analyze data of DOC patients gathered from PSG by some researchers [14,56,62]. However, all kinds of brain damage that may cause relatively similar clinical manifestations of an unconscious state may have many differences in how they alter brain activities and the sleep patterns observed as a result. Therefore, we suggest that revising and updating these scoring criteria is necessary if these criteria will be used for differential diagnosis or even forecasting in DOC states.

### 6.3. Environmental Factor

Notably, in research on sleep in DOC, there is an inherent problem; that is, sleep is often uncontrolled during the daytime or the stage when the light level is similar over the day-night, which may lead to a decrease in the amount of sleep recorded and evaluated at night. In addition to brain injury, the hospital environment itself is also the cause of a considerable number of sleep interruptions, which include frequent arousals, awakenings, or enhanced sleep fragmentation. Noise, light, mechanical ventilation, and nursing behavior are all factors that affect ICU patients’ sleep [51]. Together, when optimal sleep is necessary for brain plasticity changes and brain recovery, which are specifically attributed to nocturnal sleep by the existing literature, these factors can lead to the deterioration of sleep quality and even severe sleep deprivation.

In summary, the literature review of sleep EEG/PSG provides insights that can help patients with DOC promote their treatment and rehabilitation from diagnosis to prognosis. Future research should include long-term EEG/PSG recordings performed in well-documented patients with DOC and circadian measures in them. Furthermore, we will develop a sleep staging system based on deep learning for patients with consciousness disorders that can monitor patients’ sleep, categorize sleep stages and regulate patients’ sleep in real time.

## Figures and Tables

**Figure 1 brainsci-11-01072-f001:**
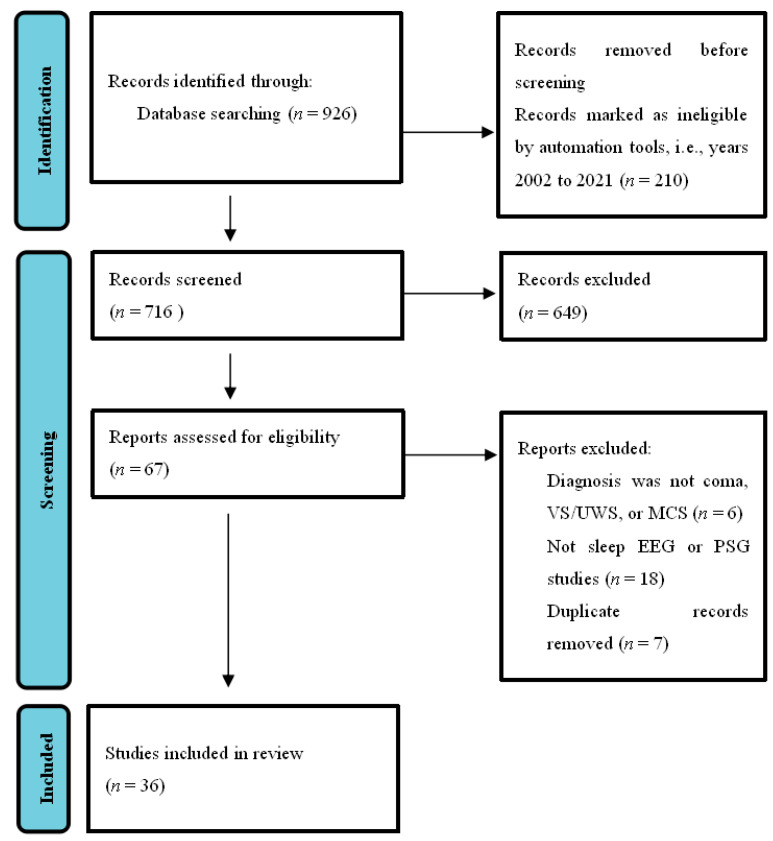
PRISMA flow chart.

**Figure 2 brainsci-11-01072-f002:**
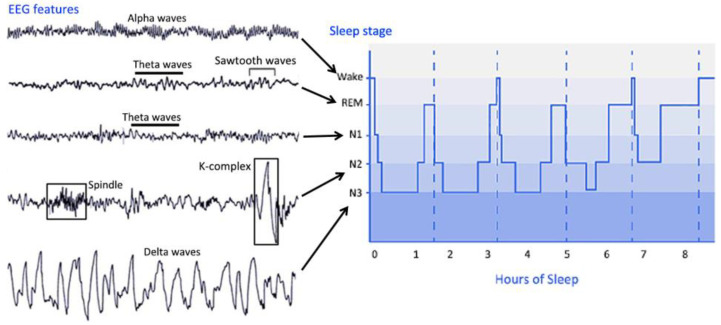
EEG features and hypnogram of sleep stages. The right side of the figure is a hypnogram, which describes the different sleep stages of 8 h of sleep at night. The EEG characteristics of each sleep stage are listed on the left. Abbreviations: REM, rapid eye movement sleep; N1, non-REM sleep stage 1; N2, non-REM sleep stage 2; N3, non-REM sleep stage 3.

**Figure 3 brainsci-11-01072-f003:**
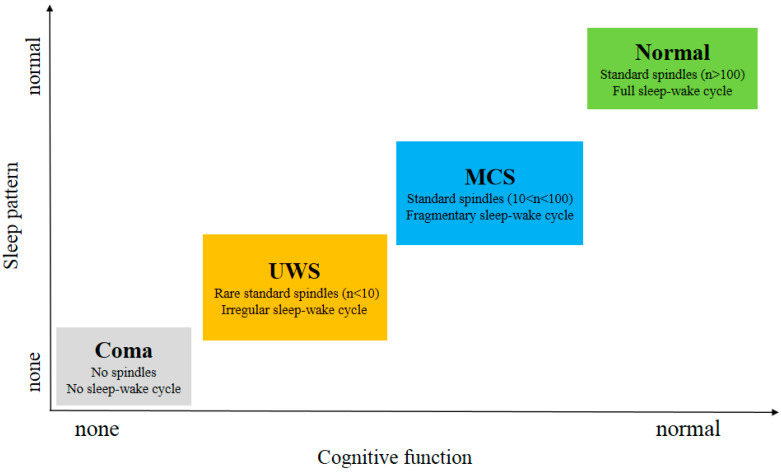
Sleep pattern and cognitive function. The awareness and cognitive function of patients with disorders of consciousness, such as UWS or MCS patients, can be assessed from sleep patterns. The sleep–wake cycle and the number of sleep spindles can reflect the state of consciousness. Abbreviations: UWS, unresponsive wakefulness syndrome; MCS, minimally conscious state.

**Figure 4 brainsci-11-01072-f004:**
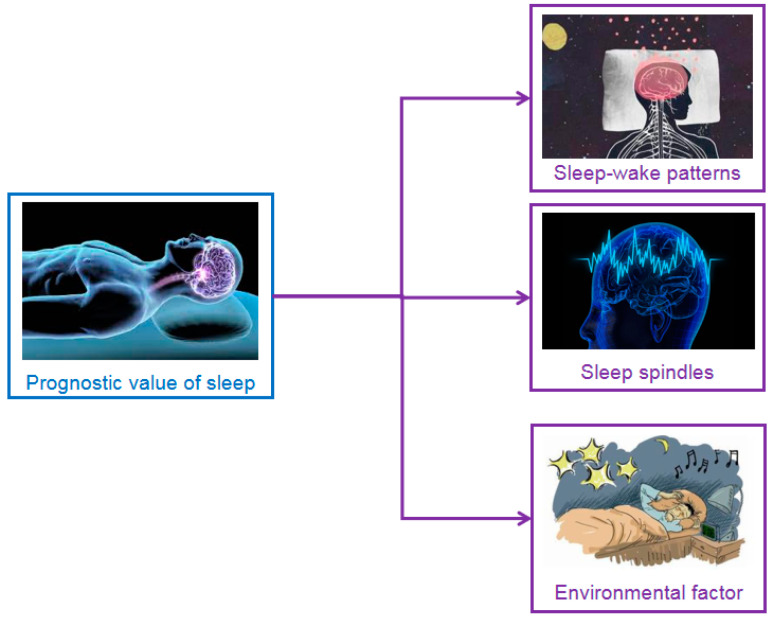
Prognosis factor of sleep for DOC patients. The prognostic value of sleep is summarized into sleep spindles, organized sleep–wake patterns and environmental factors.

**Table 1 brainsci-11-01072-t001:** Sleep phenomena in UWS and MCS patients.

Reference	N (UWS/MCS)	Sleep–Wake Cycle	SWS	REM	Spindles	Main Results
Landsness et al. (2011) [2]	5/6	5/5 UWS 6/6 MCS	not reported	0/5 UWS 5/6 MCS	0/5 UWS 6/6 MCS	MCS showed an alternating sleep pattern; UWS preserved behavioral sleep but no sleep EEG patterns;
Cologan et al. (2013) [1]	10/10	3/10 UWS 5/10 MCS	4/10 UWS 7/10 MCS	3/10 UWS 9/10 MCS	4/10 UWS 6/10 MCS	The presence of rest periods did not always indicate retention electrophysiological sleep–wake cycles that should no longer be used to differentiate UWS from MCS
Forgacs et al. (2014) [40]	8/23	5/8 UWS 22/23 MCS	2/8 UWS 13/23 MCS	2/8 UWS 9/23 MCS	4/8 UWS 18/23 MCS	EEG was well organized in patients with evidence of concealed command-following; Preservation of specific EEG characteristic could be used to differentiate UWS from MCS;
De Biase et al. (2014) [14]	27/5	22/27 UWS 5/5 MCS	not reported	4/27 UWS 5/5 MCS	15/27 UWS 5/5 MCS	The concomitant presence of sleep spindles and REM sleep correlated with patients diagnosis
Aricò et al. (2015) [45]	8/6	5/8 UWS 6/6 MCS	not reported	2/8 UWS 5/6 MCS	1/8 UWS 4/6 MCS	MCS showed more preserved sleep pattern, preserved NREM/REM sleep distribution, and physiologic hypnic figures than UWS
Arnaldi et al. (2016) [19]	20/6	17/20 UWS 6/6 MCS	not reported	5/20 UWS 3/6 MCS	17/20 UWS 6/6 MCS	The boundaries between UWS and MCS were elusive
Sebastiano et al. (2018) [46]	55/31	not reported	16/55 UWS 31/31 MCS	23/55 UWS 21/31 MCS	5/55 UWS 8/31 MCS	The presence of SWS was the most appropriate factor to differentiate UWS from MCS
Gibson et al. (2020) [47]	8/3	8/8 UWS 3/3 MCS	4/8 UWS 3/3 MCS	5/8 UWS 3/3 MCS	4/8 UWS 1/3 MCS	MCS tended to exhibit more preserved sleep pattern than UWS

Abbreviations: N, the number of patients with UWS or MCS; UWS, unresponsive wakefulness syndrome; MCS, minimally conscious state; SWS, slow-wave sleep; REM, rapid eye movement sleep.

**Table 2 brainsci-11-01072-t002:** Possible relationship of sleep activity to patient outcome.

Reference	N (UWS/MCS)	Follow Up, Months	Methods	Prognostic Factors	Main Results
Valente et al. (2002) [62]	19/5	12–34	24-h PSG	The presence of organized sleep patterns	Organized sleep patterns can predict favorable outcomes more accurately than GCS, age and neuroimaging
Alekseeva et al. (2010) [63]	64/0	2	EEG, 24-h PSG	General sleep patterns	Preserved sleep patterns were more observed in the patients with a good outcome than in the patients with a poor outcome
Landsness et al. (2011) [2]	6/5	12	EEG, PSG	Sleep patterns, sleep cycles, spindles, homoeostatic regulation of slow-wave activity	Homoeostatic regulation of slow-wave activity might be a reliable feature that predicts positive outcomes
Cologan et al. (2013) [1]	10/10	6	EEG, 24-h PSG	Sleep–wake cycles, standard sleep stages, spindles	Sleep spindles were found more in patients who clinically improved within 6 months
Forgacs et al. (2014) [40]	8/23	6	EEG	EEG background	The overall brain metabolism of subjects with severely abnormal EEG background is significantly lower than those with normal/mildly abnormal or moderately abnormal EEG background
De Biase et al. (2014) [14]	27/5	3–144	24-h PSG	Sleep–wake cycles, spindles and REM sleep	The integrity of the preservation of sleep elements (sleep–wake cycle, sleep spindles, K-complexes, and REM sleep) is often positively correlated with clinical scores
Kang et al. (2014) [64]	56/0	12	PSG	Motor response, type of BI, EEG reactivity, spindles and N20	Motor response, type of BI, EEG reactivity, sleep spindles and N20 are important factors in predicting the recovery of awareness
Avantaggiato et al. (2015) [65]	27/0	36	14-h PSG	The presence of an organized sleep pattern, REM sleep, spindles	In the subacute stage, the presence of organized sleep patterns, REM sleep and sleep spindles often predict more favorable outcomes
Arnaldi et al. (2016) [19]	20/6	6–38	24-h PSG	Persistent and more organized sleep patterns	Sleep patterns were valuable predictors of a favorable outcome in subacute patients
Wislowska et al. (2017) [23]	18/17	1–150	24-h PSG	Density of slow waves and spindles	The density of slow waves and sleep spindles was a reliable prognostic factors
Sebastianoet al. (2018) [46]	55/31	25	24-h PSG	The presence of NREM sleep, SWS	The existence of NREM sleep (namely, SWS) reflects that the circuits and structures required for DOC patients to maintain this stage of sleep are better protected
Gibson et al. (2020) [47]	8/3	not reported	24-h PSG	Sleep microarchitecture	Sleep microarchitecture can help delineate the nature and consequences of severe acquired brain injury and provide complimentary insight into the primary and secondary symptoms of the DOC

Abbreviations: N, the number of patients with UWS or MCS; UWS, unresponsive wakefulness syndrome; MCS, minimally conscious state; PSG, polysomnography; EEG, electroencephalogram; REM, rapid eye movement sleep; SWS, slow-wave sleep.

## Data Availability

The data presented in this study are available on request from the corresponding author.

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
