# Peer review of "A Systematic Review of Sleep in Patients with Disorders of Consciousness: From Diagnosis to Prognosis"

_brainsci, 2021, doi:10.3390/brainsci11081072_

Round 1
Reviewer 1 Report
The authors of the article “A Review of Sleep in Patients with Disorders of Consciousness: From Diagnosis to Prognosis“ analyzed the concepts of sleep patterns in patients suffering from DOC, identification of this challenging population, and the prognostic value of sleep. For this they searched the literature on patients in unresponsive wakefulness state (UWS) and minimally conscious state (MCS) after severe traumatic or nontraumatic brain injury.
They showed that most MCS patients had sleep and wake alternations, sleep spindles and rapid eye movement (REM) sleep, while UWS patients have few EEG changes.
This article is well written and very important, however there are major and minor revisions:
Page 1, line 31: The term “vegetative state (VS)” is obsolete! Only use “unresponsive wakefulness state (UWS)”.
The concept of ‘vegetative state’, a term coined back in the sixties, has since been transformed and replaced by other terms with a less negative connotation, such as ‘unresponsive wakefulness syndrome’. In parallel, new clinical categories (minimally conscious state or minimally conscious plus) have appeared since it has been acknowledged that there are patients with a low level of consciousness but who nevertheless show signs that are consistent with interaction with the environment by means of unmistakeably voluntary behaviours in response to orders or gestures.
Page 2, line 56: „inked“ – do you mean “linked”
Page 2, line 65 +: This should be in the methods!
This systematic review was carried out in accordance with PRISMA guidelines [1]. The PubMed database using the concepts of sleep and DOC. Search words included ((disorders of consciousness) OR (DOC)) AND (sleep)), and the field search was (title/abstract). We emphasized the articles published in the last 20 years regarding the diagnosis and classification of DOC, and only articles with patients diagnosed with coma, VS/UWS, or MCS were included. Suggestion: „No language restrictions were applied. All types of studies were considered but only studies presenting original data were included in downstream analyses. Additionally, reference lists of included articles were also followed up to check for additional relevant studies that might have been missed.“
A PRISMA- graph would be useful!
- Liberati A, Altman DG, Tetzlaff J, et al. The PRISMA statement for reporting systematic reviews and meta-analyses of studies that evaluate healthcare interventions: explanation and elaboration. BMJ. 2009;339: b2700.
Page 2, line 69: “The number of journal papers found from 2002-2021 was 709.” In the references there are only 48 articles cited! What happened to the rest?
Page 5, line 183: “Several studies…“ – no references!
Page 6, lines 224 – 232: Irrelevant information- shorten!
Page 6, line 232: usinf the abbreviation REM is correct, I would prefer to write “rapid eye movements”
Table 1 uses “UWS” and “VS”: Only use “UWS” (also in the text).
The Chapters 2.1 and 2.2 are too long and should be shortened!
By the way R&K had REM Sleep, sleep stages “S1 – S4” (Not “N1 – N4”)!
Author Response
The authors are grateful to the first reviewer for the insightful comments and constructive suggestions. In light of your comments and suggestions, the paper has been revised. Please see our point to point responses in the attached file.

Reviewer 2 Report
In my personal experience, using the Neurowave is helpful. Neurowave is an innovative integrated system for the application of multisensory stimulation. It is used for the acquisition and analysis of biophysiological parameters of patients in VS and MCS. It is an instrument of easy application, small in size and easily transportable in the intensive care unit. The analysis produced by the system are multiple: VEP evoked potentials from flashes and patterns; event-related potentials (P300, MMN, N400, P600...), with review of individual events; EEG spectral reactivity to structured and unstructured stimulations; EEG reactivity to Intermittent Light Stimulus (SLI); EEG data clustering, to verify the presence or absence of multiple EEG stages in the recording and their characteristics; extensive features to clean the EEG signal from artifacts (Independent Component Analysis, regression from EOG); reactivity of Heart Rate Variability in front of emotigenic stimulations.
Author Response
The authors are grateful to the second reviewer for the insightful comments and constructive suggestions. In light of your comments and suggestions, the paper has been revised. Please see our point to point responses in the attached file.

Reviewer 3 Report
The review by Pan and colleagues is dedicated to the analysis of sleep patterns in chronic DOC patients and provides us with the broad overview of the role of sleep studies in this patient population. This review may help to get acquainted with the problem, however, it seems to be somewhat vague and inconsistent.
E.g., the review starts with the EEG-based defnition of normal sleep stages (with no mention of phisiological background for them, but they will be discussed scarcely in the following sections), that is followed by the 2.2 Methods section, mentioning PSG and authors own experience in automatic sleep staging (lines 141-151), that seems a bit isolated as no other methods for sleep stages detection are descibed in detail here. Next section, 2.3 Sleep stages in patients with DOC, is dedicated not to abnormaties of sleep stages in DOC patients but to approaches to automated detection of sleep architecture in DOC. The former are presented in the next section (3. Diagnosis of patients with DOC using sleep EEG) as a figure 2 and in Section 3.3 The difference between a VS and an MCS. Also, again NREM and REM stages of sleep are discussed in here (lack of references shoud be noted). The next section on prognosric factors related to sleep patterns seems more structured.
In general, the review might benefit greatly if re-arranged and presented in more conscise and structured way.
Further spell checking and English proof-reading might be suggested to increase the readability of the paper.
Author Response
The authors are grateful to the third reviewer for the insightful comments and constructive suggestions. In light of your comments and suggestions, the paper has been revised. Please see our point to point responses in the attached file.

Round 2
Reviewer 1 Report
The authors have provided a revised version. The manuscript has been sufficiently improved to warrant publication in Brain Sciences, however there is still a minor correction necessary.
Page 3 figure 1.: The figure shows that 649 from 716 studies were excluded (this is over 90%!), however it is not clear, why.
This must be described better for a systemic review.
The title should be:
-
A Systematic Review of Sleep in Patients with Disorders of Consciousness: From Diagnosis to Prognosis
Author Response
The authors are grateful to the third reviewer for the insightful comments and constructive suggestions. In light of your comments and suggestions, the paper has been revised. Please see our point-by-point responses in the attached file.

Reviewer 3 Report
The paper improved after revision. Major issues have been addressed. However, language still has to be improved (e.g., "colleges" in line 287, "only the stimulation-induced vibration EEG responses are importance" in line 314, etc.).
Author Response

(The authors gave the same response as above.)
